# Characterization of Microchannel Replicability of Injection Molded Electrophoresis Microfluidic Chips

**DOI:** 10.3390/polym11040608

**Published:** 2019-04-02

**Authors:** Bingyan Jiang, Laiyu Zhu, Liping Min, Xianglin Li, Zhanyu Zhai, Dietmar Drummer

**Affiliations:** 1State Key Laboratory of High Performance Complex Manufacturing, College of Mechanical and Electrical Engineering, Central South University, Changsha 410083, Hunan, China; jby@csu.edu.cn (B.J.); zly_csu@163.com (L.Z.); minliping@csu.edu.cn (L.M.); lixianglin@csu.edu.cn (X.L.); 2Institute of Polymer Technology (LKT), University Erlangen-Nuernberg, Am Weichselgarten 9, 91058 Erlangen, Germany; drummer@lkt.uni-erlangen.de

**Keywords:** injection molding, microfluidic chip, replicability accuracy, microchannel

## Abstract

Microfluidic chips have been widely applied in biochemical analysis, DNA sequencing, and disease diagnosis due to their advantages of miniaturization, low consumption, rapid analysis, and automation. Injection molded microfluidic chips have attracted great attention because of their short processing time, low cost, and mass production. The microchannel is the critical element of a microfluidic chip, and thus the microchannel replicability directly affects the performance of the microfluidic chip. In the current paper, a new method is proposed to evaluate the replicability of the microchannel profile via the root mean square value of the actual profile curve and the ideal profile curve of the microchannel. To investigate the effects of injection molding parameters (i.e., mold temperature, melting temperature, holding pressure, holding time, and injection rate) on microchannel replicability, a series of single-factor experiments were carried out. The results showed that, within the investigated experimental range, the increase of mold temperature, melt temperature, holding pressure, holding time, and injection rate could improve microchannel replicability accuracy. Specifically, the microchannels along the flow direction of the polymer melt were significantly affected by the mold temperature and melt temperature. Moreover, the replicability of the microchannel was influenced by the distance from the injection gate. The effect of microchannel replication on electrophoresis was demonstrated by a protein electrophoresis experiment.

## 1. Introduction

The microfluidic chip is a technology platform that integrates sample preparation, reaction, separation, detection, and other processes by controlling fluid flow in microchannels [1,2,3]. Since the first demonstration in 1990 [4,5], microfluidic chip electrophoresis (MCE) has developed rapidly with important applications in many areas, such as life sciences, biology, medicine, food, and environmental monitoring [6,7,8,9,10]. This method for protein analysis is automated, which allows for rapid separation, and high sensitivities and efficiencies [11,12,13,14,15].

There are currently a number of techniques to produce microfluidic devices, including wet etching, reactive ion etching, conventional machining, soft lithography, hot embossing, injection molding, laser ablation, in situ construction, and plasma etching [16]. Injection molding technology has attracted great attention because of its short processing time, low cost, and mass production ability [17]. However, due to the micro- or even nanometer size of the microchannel, the scale-dependent convective heat transfer and viscous dissipation between the polymer melt and the mold cavity surface affect polymer melting during the injection molding process. Moreover, because of the scale-dependent effect of the flow behavior, it is difficult for polymer melt to fill into the mold cavity. Thus, low microchannel replicability accuracy and differing degrees of microchannel replication at different positions in the same chip can be found [18]. During the biochemical analysis, the pressure drop in the microchannel is greatly affected by its dimensions [19]. The microchannel replication directly affects the electric field intensity and the distribution of the electro-osmotic flow field in the microchannel, which in turn affects the efficiency of chip electrophoresis [20]. Therefore, it is very important to investigate the replication accuracy of microfluidic microchannels.

The replication accuracy of microfluidic microchannels is defined by the similarity between the actual size of the microchannel and the mold design size during the injection molding. Loke et al. [21] found that injection molding technology can provide a high replicability accuracy in the overall structure of the microfluidic chip, but that there are some defects in the microchannel, such as incomplete filling. Utko et al. [22] introduced deoxyribonucleic acid (DNA) tensile experiments to improve the performance of injection molded microfluidic chips. Lee [23] used a modular and segmented micro-mold system to form a semi-circular, cross-section microfluidic chip. The formed chip can better replicate the shape of the core, the surface of the microchannel is smooth, and the dimensional accuracy is high. Yang et al. [24] used orthogonal experiments to establish the relationship between injection process parameters and replication accuracy for various microchannels. Experimental results showed that the mold temperature and the packing pressure are the principal factors in the molding process. Song et al. [25] characterized microchannel replication based on the microchannel opening width and studied the reasons for the occurrence of the opening width. It was demonstrated that the mold temperature had a significant influence on microstructural replication. Fu et al. [26] studied the effect of process parameters on the replication accuracy of microchannels by measuring the change in the middle width of the microchannel, and provided the optimized process parameters. However, at present, most researchers only use the depth or width dimension of microchannels as an indicator to evaluate microchannel replication, which neglects the shape of the two-dimensional, cross-sectional profile of the microchannel. The depth or width dimension of microchannels cannot give a full description of the microchannel features.

In order to solve the above problems, a new method for characterizing the microchannel replication, via the root mean square value of the actual profile curve and the ideal profile curve of microchannel, is proposed in the present paper. In addition, to achieve high replication accuracy, the injection process parameters for the microfluidic chip were optimized. An experimental validation using protein electrophoresis was also carried out.

## 2. Experiments

### 2.1. The Dimension of the Microfluidic Chips

A microfluidic electrophoresis chip with cross-shaped microchannels was used in this paper. The cross-shaped microfluidic electrophoresis chip is a relatively classic commercial chip that is widely used in laboratory research. The chip is composed of a substrate and a cover sheet. The substrate contains cross-microchannels, reservoirs, and other structures. The dimensions of the chip are 5.0 cm × 2.8 cm × 0.8 cm. The cover sheet is an ordinary flat part with a thickness of 0.6 mm. The overall structure of the substrate and the cross-sectional dimensions of the microchannel are shown in Figure 1.

### 2.2. Experimental Materials and Molding Equipment

Polymethylmethacrylate (PMMA; Chimei CM-205, Taiwan) was used as the chip material, and its properties are shown in Table 1. This material has excellent optical properties, electrochemical properties, and biocompatibility, which fully meet the requirements of the chip in terms of material optics and processability. Injection molding technology was employed to fabricate the microfluidic chips. The double-cavity mold was designed to produce the substrate and cover sheets at one time, which can significantly shorten the processing time and reduce the cost of microfluidic chips.

The mold insert was prepared by wet etching and precision electroforming with a thickness of 3 mm. The experimental mold and mold insert of the substrate are shown in Figure 2. The injection molding machine used was the Arburg 370S precision injection molding machine from Lossburg, Germany. A mold temperature controller (Shini, STM-607, Dongguan, China) was used to control the mold temperature.

### 2.3. Injection Molding Experimental Design

Using a single-factor experimental method, the influence of mold temperature, melt temperature, holding pressure, holding time, and injection rate on microchannel replicability were studied. The values for each experimental factor are shown in Table 2. The material was dried at 80 °C for 8 h using a dry feeder (Shini SCD-20μ/30H, Dongguan, China) prior to the experiment. The waiting time is needed to guarantee the stability of samples after changing any of the process parameters. Five samples were produced for each set of process parameters. Microchannel replicability tests were conducted after 24 h at a constant temperature (23 °C) and humidity (50%).

### 2.4. The Evaluation of Microchannel Replicability

The microchannel structure of the substrate sheet was measured using a laser confocal microscope (Zeiss Axio LSM700, Jena, Germany). The detection positions can be found in Figure 3. Position A was located on the vertical microchannel, 3 mm from the intersection with the lateral microchannel. Positions B, C, and D were located on the lateral microchannel, 10, 20, and 30 mm away from the injection gate, respectively. The four detection positions on each sample were sequentially inspected using a magnification of 500 times.

The root mean square (RMS) was used to evaluate the deviation between the ideal profile and the actual profile of the microchannel. Compared with the parameters of the microchannel width and area, the RMS value could reflect microchannel replication more accurately. A low RMS value means a better replication accuracy. As shown in Figure 4, peak to valley (PV) is the measurement of the height difference between the highest and lowest points on the actual curve. Equation (1), which was used for the calculation of the RMS value, is given below.
(1)$$RMS=\sqrt{\frac{{\left(N_{1}-n_{1}\right)}^{2}+{\left(N_{2}-n_{2}\right)}^{2}+\ldots +{\left(N_{400}-n_{400}\right)}^{2}}{400}},$$
where *n* is a point on the actual profile curve and *N* is the corresponding point on the ideal profile curve.

In this study, 400 data points in the microchannel actual profile curve were selected. The MATLAB program was used to calculate the RMS values of the actual profile curve and the ideal profile curve. To be specific, the MATLAB program was used to complete the following: draw the actual profile curve using 400 data points, determine the symmetry axis of the actual profile curve, draw the ideal profile curve using the axis of symmetry as a reference, and calculate the RMS value of the 400 data points on the actual profile curve from the ideal profile curve.

### 2.5. Protein Electrophoresis Experiment

Thermo Scientific PageRuler Prestained Protein Ladder (10 to 180 kDa, Thermo Scientific, Lot. 26616, Waltham, MA, USA) was used in protein electrophoresis. Prior to electrophoresis runs, each microchannel was filled with an acrylamide gel solution. Table 3 gives the quantities used to prepare the 12% acrylamide gel concentrations used in this work. After 30 min, the gel was polymerized inside the microchannels. An SDS-Tris-glycine buffer (pH 8.6) was injected into the microchannels filled with gel. The channels were equilibrated by applying a 100 V charge between electrodes S and SW for 5 min. After that, a 600 V charge was applied between electrodes B and BW for another 5 min (see Figure 1). Afterwards, the buffer was removed from reservoir S and 10 μL of the Prestained Protein Ladder sample was loaded on it. Subsequently, an injection voltage of 100 V was applied between electrodes S and SW to fill the intersection of the cross channels with the sample. Finally, a separation voltage of 600 V (200 V/cm) was applied between electrodes B and BW and the proteins were allowed to migrate along the horizontal channel for 1 min. The microchannel was placed under a microscope light and images were collected using light microscopy.

## 3. Results and Discussion

### 3.1. Determination of Microchannel Replication

The laser confocal micrograph of the microchannel is shown in Figure 5. The actual profile curve and ideal profile curve calculated by MATLAB are shown in Figure 6. Correspondingly, the microchannel replication was calculated according to Equation (1).

### 3.2. Influence of Process Parameters on Microchannel Replication

#### 3.2.1. The Effect of Mold Temperature on Microchannel Replication

As described in Table 1, the molding temperature was varied in the range of 60 to 100 °C while the other process parameters were kept at the standard level. Figure 7 shows the effect of the mold temperature on the microchannel replicability.

The RMS value at each detection position decreased as the temperature of the mold increased. The replicability of the microchannel was unresponsive to the mold temperature. The large temperature difference between the molten polymer and the mold led to a thick condensing layer during the melt flow, which increased the filling resistance. Therefore, in this case, the microchannel replicability was generally poor.

When the mold temperature was increased from 60 to 80 °C, the filling resistance of the polymer melt gradually decreased and, thus it easily filled the cavity. Correspondingly, the RMS values at Positions B, C, and D were reduced by 1.62, 2.05, and 1.50 μm, respectively, and the microchannel replicability was significantly improved. Compared with the mold temperature of 60 °C, the RMS at 80 °C was reduced by 24.31%, 26.64%, and 18.82% for positions B, C, and D, respectively, and the microchannel replication was significantly improved. However, the RMS at position A of the longitudinal microchannel only decreased by 1.13 μm. This is because when the melt flows through the micro-embossed structure at core A, it will flow in the filling direction with low resistance. The hysteresis effect caused by the polymer melt meant that the filling at the near-gate side at position A was smaller than at the far-gate side. When further increasing the mold temperature, the change in RMS values of the microchannel tended to be less, so a mold temperature of 80 °C is suggested.

#### 3.2.2. Effect of Melt Temperature on Microchannel Replicability

The melt temperature was varied in the range of 230 to 270 °C while the other process parameters were kept at the standard level. Figure 8 shows the effect of the melt temperature on the microchannel replicability.

With the increase of the melt temperature, the RMS value of each detection position first decreased, and then increased. When the melt temperature was low, the polymer melt had a high viscosity and it was difficult to fill in the cavity. The RMS value of each detection position was greater than 7.12 μm. The increase of the melt temperature improved the fluidity of the polymer melt, and reduced the pressure and temperature loss during the injection process. When the melt temperature was increased to 250 °C, the RMS values of detection positions A, B, C, and D decreased by 1.06, 2.07, 1.66, and 1.32 μm, respectively. As the melt temperature was increased, the RMS value increased. An excessive temperature difference between the melt and the mold temperature causes the surface of the microfluidic chip to have a large residual stress. The stress release process causes the microchannel to deform, so the melt temperature should not exceed 250 °C.

#### 3.2.3. Effect of Holding Pressure on Microchannel Replicability

As described in Table 1, the holding pressure was varied in the range of 80 to 120 MPa while the other process parameters were kept at the standard level. Figure 9 shows the effect of the holding pressure on the microchannel replicability.

The microchannel replicability at each detection position increased with the holding pressure. When the holding pressure was 80 Mpa, a compensating shrinkage of the melt was not obvious after the cavity was filled. Especially for the longitudinal microchannel, the lower holding pressure could not effectively counteract the hysteresis effect, so there was obvious asymmetric roundness at position A. Correspondingly, the RMS value here was found to be 8.27 μm. As the holding pressure increased, the back pressure also increased after the melt was filled. Thus, the melt could be pressed into the gap between the part and the core, so that the microchannel size was closer to the designed size. In general, the effect of holding pressure on the longitudinal microchannel was significantly greater than that on the lateral microchannel.

#### 3.2.4. Effect of Holding Time on Microchannel Replicability

As described in Table 1, the holding time was varied in the range of 1 to 5 s and the other process parameters were kept at the standard level. Figure 10 shows the effect of the holding time on the microchannel replicability.

Increasing the holding time can increase the back pressure of the cavity at the end of the filling. Furthermore, it can strengthen the compensating shrinkage and reduce the incompleteness of the microchannel filling to increase the replicability. When the holding time was increased from 1 to 3 s, the RMS values of the microchannel at positions A, B, C, and D decreased by 1.63, 1.49, 1.91, and 1.70 μm, respectively. When the holding time exceeded 3 s, the RMS value of the transverse microchannel tended to remain stable. The effect of the holding time on the longitudinal microstructure was greater than that on the transverse microstructure.

#### 3.2.5. Effect of Injection Rate on Microchannel Replicability

As described in Table 1, the injection rate was varied in the range of 20 to 40 cm^3^/s and the other process parameters were kept at the standard level. Figure 11 shows the effect of the injection rate on the microchannel replicability.

As the injection rate increased, the root mean square difference of each detection position decreased and the microchannel replication was improved. When the injection rate was 20 cm^3^/s, the melt filling was slower. There was a high shearing force and a large filling resistance. The RMS of each point was greater than 7.20 μm. Increasing the injection rate shortened the filling time, enhanced the melt shearing action, and kept the melt filling the cavity at a higher temperature. When the injection rate was increased from 20 to 30 cm^3^/s, the RMS of longitudinal microchannel positions B, C, and D decreased by 2.15, 1.87, and 1.82 μm, respectively. When the injection rate was increased to 40 cm^3^/s, the change in RMS tended to lessen. The lateral microchannel position A was significantly affected by the injection rate. When the injection rate was 40 cm^3^/s, the RMS was 20% lower than that at 20 cm^3^/s. It can be seen that the injection rate had a greater influence on the lateral microchannel than on the longitudinal microchannel.

### 3.3. Protein Electrophoresis

In order to compare the effect of microchannel replication on the quality of electrophoresis, microfluidic chips produced at the melt temperature of 230, 250, and 270 °C were selected. The corresponding results are presented in Figure 12. It can be seen that the Protein Ladder achieved a good separation in the microfluidic chip at the melt temperature of 250 °C. Proteins with different molecular mass (kDa) formed clear ladders. However, the protein in the microfluidic chip at the melt temperatures of 230 and 270 °C failed to separate effectively. Among them, the separation effect of 230 °C was the worst, and the protein could not form a uniform band. In this case, the RMS value of each detection position was greater than 7.12 μm. When a certain voltage was applied across the microchannel, a uniform electric field could not be formed inside the microchannel. It was demonstrated that the degree of replication of the microchannels has a significant effect on the electrophoresis results. Follow-up research will be designed to investigate the influence factors of the electric field distribution and electrophoresis performance.

## 4. Conclusions

To give an accurate description of microchannel replicability, a new method was proposed in which the RMS value of the actual profile of the microchannel was employed. By means of this method, it was found that increases in mold temperature, melt temperature, holding pressure, holding time, and injection rate could increase the microchannel replication accuracy. The longitudinal microchannels were significantly affected by the mold temperature and the melt temperature, while the transverse microchannels were most affected by the holding time. As the distance from the injection gate increased, the vertical microchannel replication accuracy decreased. The hysteresis effect on the transverse microchannel was obvious. The transverse microchannel replication accuracy was generally lower than that of the longitudinal microchannel. The degree of replication of the microchannels had a significant effect on the electrophoresis results.

## Figures and Tables

**Figure 1 polymers-11-00608-f001:**
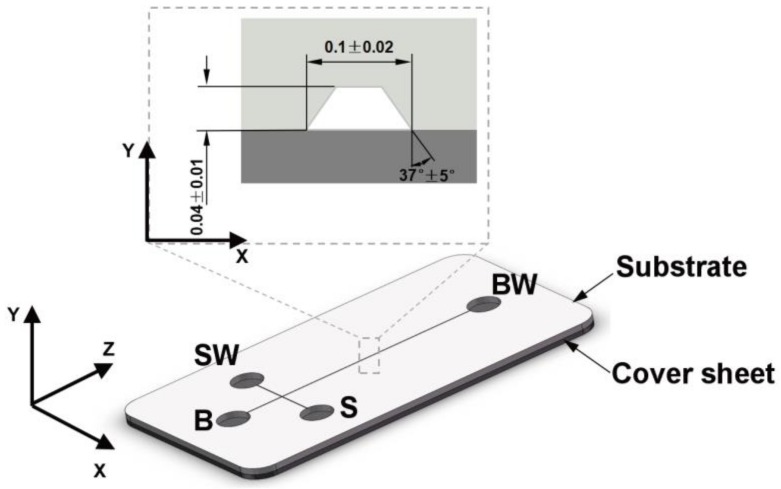
The structure of the microfluidic chip (Unit: mm). B, S, BW, and SW denote reservoirs containing buffer, sample, buffer waste, and sample waste, respectively.

**Figure 2 polymers-11-00608-f002:**
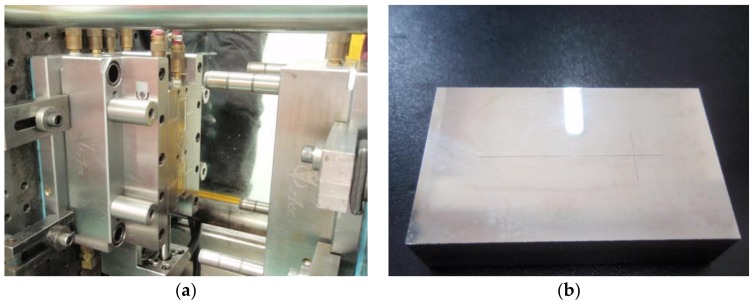
(**a**) The experimental mold of the microfluidic chip and (**b**) The mold insert of the substrate.

**Figure 3 polymers-11-00608-f003:**
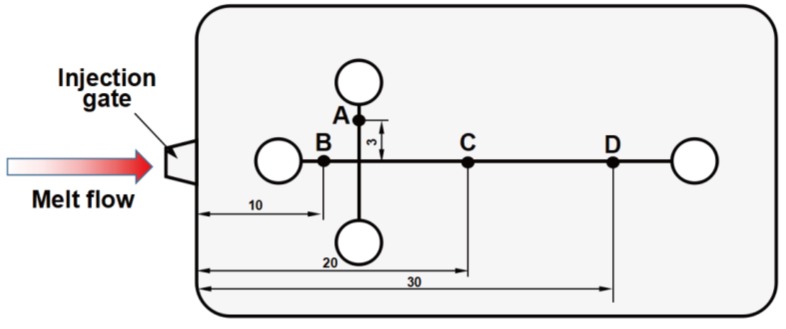
The distribution of the detection positions (A, B, C, D) in the substrate sheet (unit: mm).

**Figure 4 polymers-11-00608-f004:**
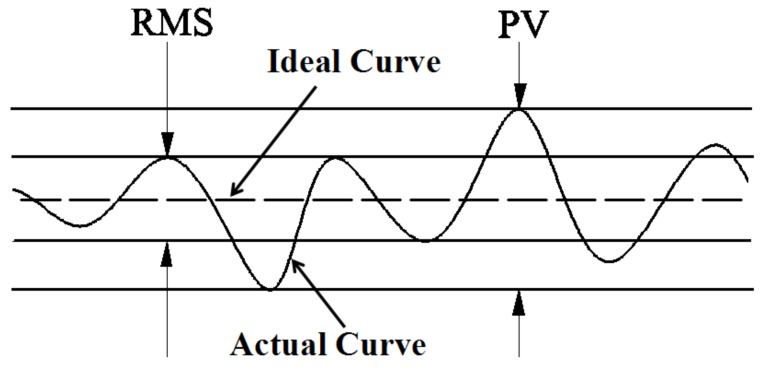
The peak to valley (PV) and the root mean square (RMS) of the actual curve and the ideal curve.

**Figure 5 polymers-11-00608-f005:**
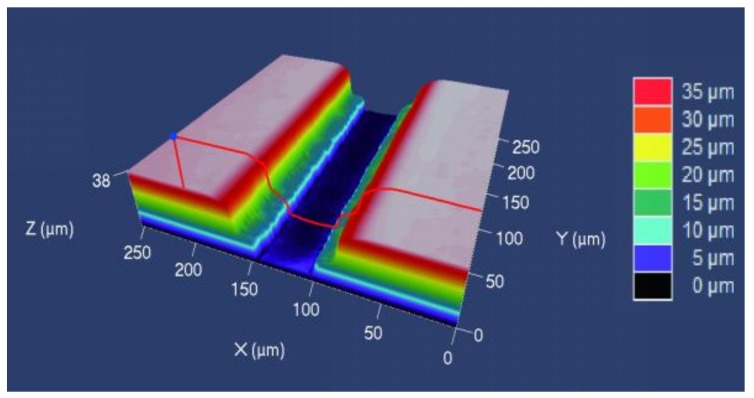
The laser confocal micrograph of the microchannel.

**Figure 6 polymers-11-00608-f006:**
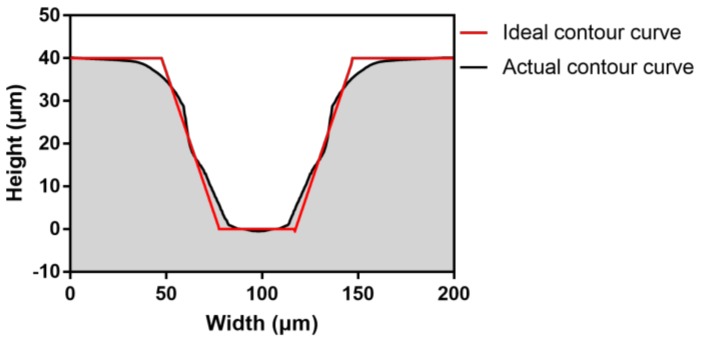
The ideal and actual contour curves of the microchannel.

**Figure 7 polymers-11-00608-f007:**
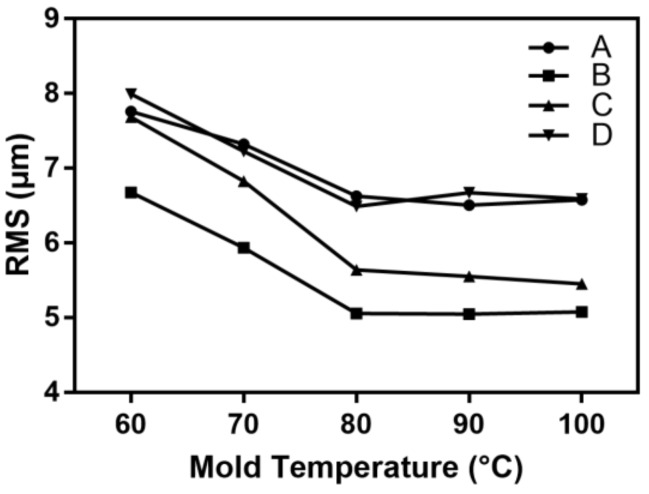
The influence of mold temperature on the RMS of the microchannel.

**Figure 8 polymers-11-00608-f008:**
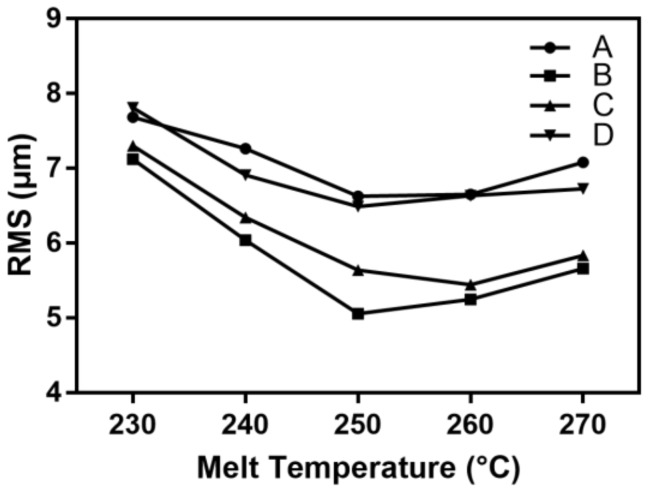
The influence of the melt temperature on the RMS of the microchannel.

**Figure 9 polymers-11-00608-f009:**
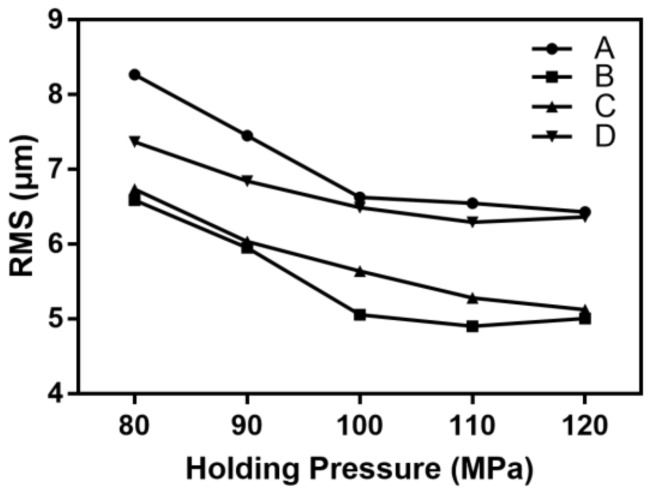
The influence of holding pressure on the RMS of the microchannel.

**Figure 10 polymers-11-00608-f010:**
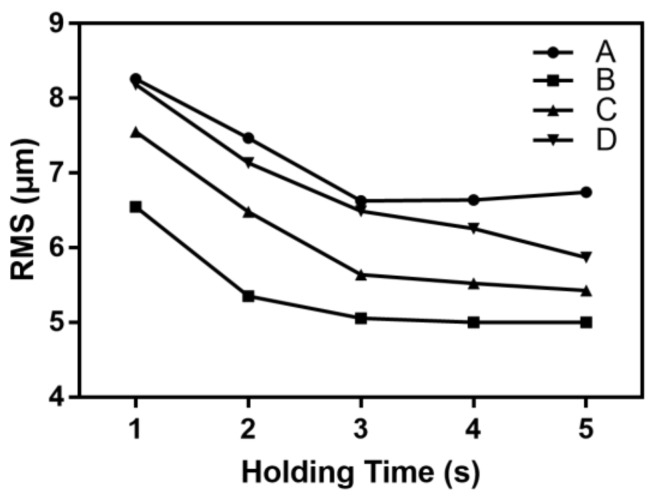
The influence of the holding time on the RMS of the microchannel.

**Figure 11 polymers-11-00608-f011:**
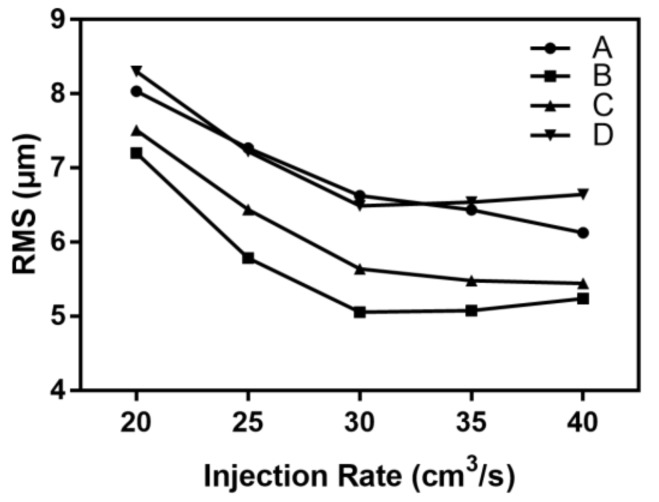
The influence of the injection rate on the RMS of the microchannel.

**Figure 12 polymers-11-00608-f012:**
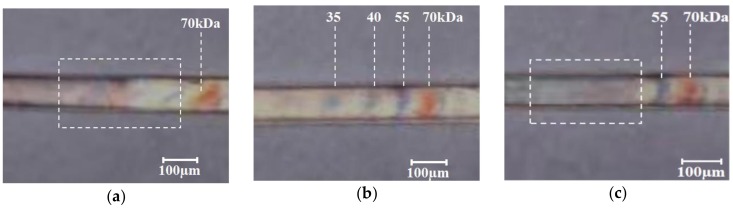
The protein in the microchannels after electrophoresis when melt temperature was: (**a**) 230 °C, (**b**) 250 °C, and (**c**) 270 °C, and the other process parameters were kept at the standard level.

**Table 1 polymers-11-00608-t001:** The properties of polymethylmethacrylate (PMMA; Chimei CM-205).

Properties	Level
Density (g/cm^3^)	1.19
Poisson’s Ratio	0.33
Thermal Conductivity (W/(m · K))	0.2
Specific Heat Capacity (J/(kg · K))	1500
Coefficient of Thermal Expansion (1/K)	7 × 10^−5^

**Table 2 polymers-11-00608-t002:** The single-factor experimental program.

Process Parameters	Level 1	Level 2	Standard	Level 3	Level 4
Mold Temperature (°C)	60	70	80	90	100
Melt Temperature (°C)	230	240	250	260	270
Holding Pressure (MPa)	80	90	100	110	120
Holding Time (s)	1	2	3	4	5
Injection Rate (cm^3^/s)	20	25	30	35	40

**Table 3 polymers-11-00608-t003:** The protocols used to prepare the gel used in protein electrophoresis.

Composition	Level
Deionized water (µL)	350
Acrylamide/bis-acrylamide 30% (µL)	400
Tris-HCl 1.5 M (µL)	250
Ammonium persulfate 10% (µL)	10
Tetramethylethylenediamine (µL)	0.83

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
