# Peer review of "Characterization of Microchannel Replicability of Injection Molded Electrophoresis Microfluidic Chips"

_polymers, 2019, doi:10.3390/polym11040608_

Round 1

Reviewer 1 Report

1.       In the abstract (lines 21-22) the authors wrote: “….the replicability of microchannel is represented by the root-mean-square value of microchannel profile.” This phrase is not clear and does not explain what the authors carried out. A good explanation is in the introduction (lines 71-72). Please modify.

2.       In the abstract (line 22) the authors wrote: “to investigate the effect of processing parameters…” At this point of abstract the injection molding process has not been yet cited, thus the word “processing” is not clear to what is referred to. The authors must briefly introduce the process before its parameters.

3.       In the abstract (line 27): the statement “lateral microchannel” is not easy to understand for abstract readers that haven’t seen the sample image. The abstract should be more exhaustive. Please clarify.

4.       Since from abstract, but especially in the introduction, the authors cited the word “replication”. This word has not an unique meaning because different authors choose different way to describe it, as you can find in literature. Thus, the authors should  define what they mean for replication. 

5.       Presenting ref. 23 line 60, the authors introduce the bonding temperature, bonding pressure and bonding time that belong to a process different from injection molding. Thus, these results are not useful for the comprehension of the problem of the paper.

6.       The authors should specify how was designed the chip. Is it a commercial design or is it from authors?

7.       Line 79  put “the” instead of “The”.

8.       Please add some information about materials properties (PMMA Chimei) for example the MVR  or density.

9.       Line 95 “substrate” instead of “substare”

10.   Line 109: in the text there are cited the positions C,D and E? In the figure 3 there are A,B,C, and D. Please verify.

11.   The authors use a single factor experimental method that does not give any information about process parameters interaction that instead could be useful for process optimization. Why did you choose this kind of plan? It should be better to design a factorial plan following DoE principles.

12.   Discussing mold temperature effects, it is important to evidence that is not useful to overcome 80°C because the RMS is more or less equal for 90 and 100 °C and it is not necessary increase further the temperature.

13.   At first impression I believe that the RMS results (fig. 9 and 10) for position C and D should be similar because their positions are quite near, instead D is similar to A and C to B? Could the authors try to explain this behaviour?

14.   The electrophoresis has been carried out only on two samples produced at 230 and 250°C. I believe that more test are required to prove the effectiveness of proposed method.

Author Response

Answer to Reviewer 1

1. In the abstract (lines 21-22) the authors wrote: “….the replicability of microchannel is represented by the root-mean-square value of microchannel profile.” This phrase is not clear and does not explain what the authors carried out. A good explanation is in the introduction (lines 71-72). Please modify. 

Answer: 

We thank the reviewer for this suggestion. Here in the revised manuscript, we have revised the manuscript according to the recommendation. (Lines 22- 23)

2. In the abstract (line 22) the authors wrote: “to investigate the effect of processing parameters…” At this point of abstract the injection molding process has not been yet cited, thus the word “processing” is not clear to what is referred to. The authors must briefly introduce the process before its parameters.

Answer:

Thank you for the comments. We have clarified the word “processing” to what is referred to, and briefly introduce the process before its parameters. (Lines 18-19, 23)

3. In the abstract (line 27): the statement “lateral microchannel” is not easy to understand for abstract readers that haven’t seen the sample image. The abstract should be more exhaustive. Please clarify. 

Answer:

We agree with your comments. The expression of Lateral microchannel has been modified to facilitate a better understanding. (Line 28)

4. Since from abstract, but especially in the introduction, the authors cited the word “replication”. This word has not an unique meaning because different authors choose different way to describe it, as you can find in literature. Thus, the authors should  define what they mean for replication.

Answer:

We agree with your comments. Different authors choose different way to describe “replication” , but the essence is the same. The definition of replication degree is the similarity between the actual size of the microchannel and the mold design size during the injection molding. The definition have been added in the revised manuscript according to the recommendation. (Lines 57-58)

5. Presenting ref. 23 line 60, the authors introduce the bonding temperature, bonding pressure and bonding time that belong to a process different from injection molding. Thus, these results are not useful for the comprehension of the problem of the paper.

Answer:

We agree with your comments. We have deleted this reference and added other relevant references. (Lines 62-67)

6. The authors should specify how was designed the chip. Is it a commercial design or is it from authors?

Answer:

Thank you for your question. The cross-electrophoresis microfluidic chip is a commercial design. Here we add a brief introduction of the microfluidic chip, and the related content has been added to the paper. (Lines 84-85)

7. Line 79  put “the” instead of “The”.

Answer:

We agree with the comments. It has been corrected according to the recommendation.

8. Please add some information about materials properties (PMMA Chimei) for example the MVR or density.

Answer:

We agree with the comments. Detailed description of the material (Table 1) has been added in the revised manuscript according to the recommendation.

9. Line 95 “substrate” instead of “substare”

Answer:

We agree with the comments. It has been corrected according to the recommendation.

10. Line 109: in the text there are cited the positions C,D and E? In the figure 3 there are A,B,C, and D. Please verify.

Answer:

We agree with the comments. It has been corrected according to the recommendation.

11. The authors use a single factor experimental method that does not give any information about process parameters interaction that instead could be useful for process optimization. Why did you choose this kind of plan? It should be better to design a factorial plan following DoE principles.

Answer:

We agree with the comments. In the present study, we did a pre-investigation that mainly focused on the effects of injection molding parameters (i.e., mold temperature, melting temperature, holding pressure, holding time and injection rate) on the microchannel replicability,and the single factor experimental method should be sufficient to draw a conclusion. Meanwhile, based on the injection molding process parameters provided by the injection molding machine, we made the experimental design which makes it easier to provide guidance for the injection molding related process to increase the stability of microfluidic chip in injection molding. And the single factor experiment can more intuitively and accurately describe the influence of individual process parameters, and facilitate the optimization of process parameters. In accordance with your recommendations, in the next study, we will focus on the interaction between different process parameters.

12. Discussing mold temperature effects, it is important to evidence that is not useful to overcome 80°C because the RMS is more or less equal for 90 and 100 °C and it is not necessary increase further the temperature.

Answer:

We agree with the comments. The mold temperature effects above 80°C have been added in the revised manuscript according to the recommendation. (Lines 182-184)

13. At first impression I believe that the RMS results (fig. 9 and 10) for position C and D should be similar because their positions are quite near, instead D is similar to A and C to B? Could the authors try to explain this behaviour?

Answer:

Thank you for the comments. The microchannel replication at different position is affected by the hysteresis effect and the distance from injection gate to the position. To be specific, the replication in the transverse microchannel is generally lower than the longitudinal one due to more obvious hysteresis effect. So the RMS of Position A is higher than Position B, C and D. The further away from the gate, the greater the melt temperature and pressure loss and the lower the microchannel replication. So the RMS of Position D> Position C>Position B.

14. The electrophoresis has been carried out only on two samples produced at 230 and 250°C. I believe that more test are required to prove the effectiveness of proposed method.

Answer:

We agree with the comments. According to your suggestion, the electrophoresis of the sample produced at 270℃was added into the manuscript. Comparing the electrophoresis effects at these three temperatures, the relationship between the degree of replication and the electrophoresis quality can be clearly obtained. And we will make follow-up research in the influence factors of the electric field distribution and electrophoresis performance. 

Reviewer 2 Report

page 8 line 217. Injection speed term has been used. It's better to use, like other parts on paper "injection rate"

Author Response

Answer to Reviewer 2

Page 8 line 217. Injection speed term has been used. It's better to use, like other parts on paper "injection rate"

Answer:We agree with the comments. It has been corrected according to the recommendation.

Reviewer 3 Report

Very nice paper.  A few improvements needed.

1) Please check your references for replication of microchannels; a few others would be useful.

2) Why did you not use variable mold temperature control for this work?

3) Please check the English again.  There are still lots of errors.

Author Response

Answer to Reviewer 3

Very nice paper. A few improvements needed.

1. Please check your references for replication of microchannels; a few others would be useful.

Answer:Thank you for your question. We have added other relevant references according to the recommendation. (Lines 62-67)

2. Why did you not use variable mold temperature control for this work?

Answer:

Many thanks for you suggestions. In this paper the variable mold temperature control technique was not taken into consideration. However, variable mold temperature control is a very practical technology in thin-wall micro-nano injection molding, which is especially important for the molding and bonding of microfluidic chips. In our current research, the thick film heating was employed to achieve the variable mold temperature.

3. Please check the English again. There are still lots of errors.

Answer:

Thanks for your careful reading and insightful suggestions. In the revised version, all detectable typos and grammatical mistakes have been corrected.

Round 2

Reviewer 1 Report

Thanks to authors for the answers. I believe that now the paper is ok.